# The Relationship between Gastroesophageal Reflux Disease and Chronic Kidney Disease

**DOI:** 10.3390/jpm13050827

**Published:** 2023-05-13

**Authors:** Xiaoliang Wang, Zachary Wright, Eva D. Patton-Tackett, Gengqing Song

**Affiliations:** 1Internal Medicine Residency Program, Joan C. Edwards School of Medicine, Marshall University, Huntington, WV 25701, USA; wangxi@marshall.edu (X.W.); pattont@marshall.edu (E.D.P.-T.); 2Joan C. Edwards School of Medicine, Marshall University, Huntington, WV 25755, USA; wright476@marshall.edu; 3Department of Gastroenterology and Hepatology, Metrohealth Medical Center, Case Western Reserve University, Cleveland, OH 44106, USA

**Keywords:** CKD, GERD, GERD complications, Barrett’s esophagus, esophageal stricture

## Abstract

Gastroesophageal reflux disease (GERD) is commonly seen in patients with chronic kidney disease (CKD), although data on the relationship between these conditions are still limited. We aimed to explore whether CKD is related to a higher prevalence of GERD and its complications. National Inpatient Sample data were used in this retrospective analysis, including 7,159,694 patients. Patients who had a diagnosis of GERD with and without CKD were compared with patients without GERD. Complications associated with GERD that were analyzed included Barrett’s esophagus and esophageal stricture. Risk factors of GERD were used for variable adjustment analysis. Different stages of CKD were evaluated in patients with and without GERD. Bivariate analyses were performed using the chi-squared test or Fisher exact test (2-tailed) for categorical variables as appropriate to assess the difference. There were significantly different demographic characteristics between GERD patients with and without CKD regarding age, sex, race, and other co-mobilities. Interestingly, a greater prevalence of GERD was seen in CKD patients (23.5%) compared to non-CKD patients (14.8%), and this increased prevalence was consistently seen in all CKD stages. CKD patients also had 1.70 higher odds of risk of having GERD compared with non-CKD after adjustment. The association between different stages of CKD and GERD showed a similar trend. Interestingly, patients with early-stage CKD were found to have a higher prevalence and odds of risk of esophageal stricture and Barrett’s esophagus than non-CKD patients. CKD is associated with a high prevalence of GERD and its complications.

## 1. Introduction

Gastroesophageal reflux disease (GERD) is a condition in which there is a backflow of stomach contents into the esophagus, producing symptoms such as regurgitation and heartburn. GERD can also be described by the presence of characteristic mucosal injury under endoscopy or abnormal esophageal acid exposure found in a reflux monitoring test [1,2]. 

Other than classic symptoms, GERD can cause extraesophageal symptoms such as hoarseness, wheezing, asthma, chest pain, and globus sensation [3]. Established risk factors for GERD include obesity, hiatal hernia, and tobacco smoking [4]. In line with the 2022 American College of Gastroenterology (ACG) guidelines, the identification of gastroesophageal reflux disease (GERD) should predominantly focus on the manifestation of common symptoms, such as heartburn and regurgitation, in conjunction with the patient’s response to targeted acid suppression therapy. The utilization of ambulatory reflux monitoring can be considered for individuals exhibiting atypical symptoms or to verify the diagnosis when acid suppression therapy is ineffective. While endoscopy is not customarily advised for determining the presence of GERD, it can be employed to investigate potential complications or alarm symptoms. The 2022 ACG guidelines underscore the significance of basing clinical diagnoses on symptoms and treatment outcomes, rather than depending solely on objective testing measures [2,3,5] In the US, GERD prevalence is estimated at anywhere between 18 and 28% [6], which results in a significantly impaired quality of life and a high insurance encumbrance. 

GERD is a complex condition influenced by a variety of risk factors. Prominent among these are age, obesity, hiatal hernia, and smoking. The incidence of GERD escalates with age, particularly in individuals over 65, where the prevalence is estimated to be between 20 and 30% [7] This is likely attributable to age-related changes in esophageal and stomach anatomy and function, as well as cumulative exposure to other risk factors. Obesity also plays a significant role in GERD development, with a higher prevalence observed in those with elevated body mass index (BMI) [8]. Obesity can contribute to GERD by increasing intra-abdominal pressure and impairing lower esophageal sphincter (LES) function, which facilitates stomach content reflux into the esophagus. Hiatal hernia is another factor associated with increased GERD risk [9]. This condition involves the stomach protruding into the chest through the diaphragm, disrupting the esophagus and LES’s typical anatomy. Consequently, stomach acid and other contents reflux into the esophagus, contributing to GERD development. Smoking is an additional risk factor, as research indicates that smokers have a higher likelihood of developing GERD compared to non-smokers [10]. Smoking may impair LES function and elevate stomach acid production. Other GERD risk factors include pregnancy, specific medications (e.g., nonsteroidal anti-inflammatory drugs and calcium channel blockers), and certain medical conditions (such as scleroderma and gastroparesis).

Chronic kidney disease (CKD) prevalence was around 14.9% in US adults based on a survey performed between 2015 and 2018, in which the prevalence of CKD among patients declined with the worsening of renal insufficiency [11]. Renal dysfunction has been related to an increased incidence of acid-related gastrointestinal dysfunction [12,13]; however, the association between GERD and chronic kidney disease (CKD) remains uncertain. Kurukawa et al. reported that GERD prevalence was 24.2% in 418 hemodialysis individuals using the questionnaire diagnosing reflux disease (QUEST) in Japan, which was significantly higher than the reported GERD prevalence in the Japanese population (16.3%) [14]. Kawaguchi et al. reported that in 156 renal failure patients who underwent endoscopic examination, the prevalence of GERD was 34% in an asymptomatic patient and increased to 44% and 50% in symptomatic and hemodialysis patients, respectively [15]. In a study involving 290 late-stage CKD (3–5) and end-stage renal disease (ESRD) requiring dialysis patients, dyspepsia, gastritis, and gastroesophageal reflux were significantly higher [16]. Abdulrahman et al. reported a significantly higher prevalence of GERD in ESRD patients (77.5%) compared to non-CKD patients (38.6%) in a small study involving 120 patients [17]. 

The causes of GERD in CKD patients are multifactorial, but the underlying mechanisms have not been thoroughly investigated. It has been suggested that uremia and electrolyte abnormality confer their effect on esophageal function. Clovis et al. found that ESRD patients requiring hemodialysis tended to have increased lower esophageal sphincter (LES) resting pressure and shorter LES relaxation time [18]. Nausea, vomiting, anorexia, and dyspepsia are prevalent complaints for ESRD patients, commonly attributed to uremia-induced impairment of gastric myoelectric activity, gastric hypomotility, and prolonged gastric emptying [19]. However, Dogu et al. showed no statistically significant differences in GERD symptoms among pre- and post-dialysis patients [16]; thus, the exact relationship between CKD and GERD remains unclear. 

This study aimed to determine if CKD has an increased association with a higher prevalence of GERD. We also aimed to assess the relationship between CKD and GERD complications, including esophageal stricture and Barrett’s esophagus.

## 2. Materials and Methods


**Database**


A retrospective analysis was performed using the 2017 National Inpatient Sample (NIS) database developed by the healthcare cost and utilization project (HCUP). NIS is the leading publicly available all-payer inpatient healthcare catalog designed to estimate inpatient utilization, access, cost, quality, and outcomes, which contains unweighted data from around seven million hospital stays each year. The NIS approximates a 20% stratified sample of all discharges from US community hospitals, not including rehabilitation and long-term acute care hospitals. 


**Data collection and outcomes**


A total of 7,159,694 adult patients admitted to hospital in 2017 were included in this study. Patients diagnosed with GERD (ICD-10-CM K21.9) with and without CKD (ICD-10-CM N18.1-9) were compared to patients without GERD. We excluded subjects with a history of upper GI surgeries, uncontrolled T2DM, eosinophilic esophagitis, and infective esophagitis. Risk factors for CKD, including controlled T2DM, essential HTN, and hyperlipidemia, and risk factors for GERD, including hiatal hernia, cigarette smoking, and obesity, were used for variable adjustment analysis [20,21]. Demographic data were collected, including age, race, and gender. GERD complications, including esophagus stricture and Barrett esophagus, were included only in patients with the diagnosis of the GERD group. To assess the odds ratio of GERD and related complications in different stages of CKD, we included case patients who had a diagnosis of GERD and associated complications compared to control patients with no diagnosis of GERD and no related complications. All diagnoses involved or omitted were selected via the ICD-10-CM code. 


**Statistical analysis**


Risk factors and demographic information from this study collected from NIS were categorical and presented as several cases and percentages. Chi-squared analysis was used to analyze the association between GERD and CKD and investigate the relationship between complications in GERD in those with and without CKD. Multivariate logistic regression analysis was used to evaluate the odds ratio—the risk of GERD and GERD complications with and without the different stages of CKD. We adjusted for gender, race, cigarette smoking, hiatal hernia, and obesity as covariates to lessen the consequence of possible confounding factors. A 2-sample test for equal proportions was used, and a *p*-value < 0.05 was considered significant. IBM SPSS 28.0.1.1 was used for the statistical analysis.

## 3. Results

A total of 7,159,694 hospitalized patients during 2017 were included in this study, among which 1,145,005 subjects diagnosed with GERD with and without CKD were identified (235,840 with CKD and 909,165 without CKD) (Figure 1). Overall, most GERD-CKD patients were older than those without CKD (71.6 ± 0.1 vs. 62.5 ± 0.1). There were significantly more female patients than male patients diagnosed with GERD, but there were considerably more male patients among the GERD-CKD patients versus the GERD patients with no CKD (47.1% vs. 39.9%, *p* < 0.01). T2DM prevalence was considerably higher in those with GERD-CKD than those without CKD (45.9% vs. 6.2%, *p* < 0.01). More patients with GERD-CKD also had hyperlipidemia (55.1% vs. 39.9% *p* < 0.01) compared to GERD without CKD. (Table 1) Among the control groups, it was observed that CKD patients without a diagnosis of GERD were significantly older than those without both CKD and GERD. Additionally, a higher proportion of female patients were present in the CKD group without GERD. The race distribution in the control groups was similar to that of the GERD-diagnosed patients. Furthermore, the prevalence of T2DM, hypertension, and hyperlipidemia was significantly higher among CKD patients without GERD than among nthose without CKD and GERD (*p* < 0.01).

Individuals with CKD were more likely to have GERD than those with no history of CKD (OR 1.70, 95% CI 1.688–1.706, *p* < 0.001). GERD incidence in those with all stages of CKD was 23.5%, and the incidence in those without CKD was 14.8% (*p* < 0.001). Interestingly, this increased incidence was consistently seen in all CKD stages and ESRD, with the highest risk (OR 1.88 95% CI 1.77–2.01, *p* < 0.001) and incidence (25.9%, *p* < 0.001) of GERD found in a patient diagnosed with stage 1 CKD. However, there is no significant clinical difference across CKD stages in the risk and incidence of GERD (Table 2 and Figure 2).

Among complications potentially associated with GERD, the risk of esophageal stricture was significantly greater in subjects with CKD (OR 1.36 95% CI 1.28–1.45, *p* < 0.001). The incidence of esophageal stricture in CKD was 17.4 per 10,000 patients, while that for those without CKD was 10.6 per 100,000 (*p* < 0.01). Interestingly, the incidence and risk of esophageal stricture were found to be higher in the early stage of CKD (CKD stage 1–3), with the highest in CKD stage 1 (OR 2.24 95% CI: 1.19–4.20 *p* = 0.01; 25.2 per 10,000 patients) and trending down as CKD stage increased. Nevertheless, there was no statistically significant difference observed in the incidence and risk of esophageal stricture, particularly in CKD stage 4 and 5 (Table 2 and Figure 2).

Barrett’s esophagus was more commonly seen in subjects with GERD-CKD than in those with no history of CKD (OR 1.23 95% CI: 1.18–1.29, *p* < 0.001), with 29.7 per 10,000 subjects among those with CKD and 19.7 per 10,000 subjects among patient without CKD (*p* < 0.01). There was a trend of a decreasing incidence of Barrett’s esophagus in GERD patients with worsening renal function, similar to that observed in esophageal stricture (CKD stage 1: 40.4 per 10,000 cases versus CKD stage 5: 23.5 per 10,000 cases). However, after adjusting for other co-variants, no statistically significant difference was found in the risk analysis in patients with CKD stage 4 and 5 (*p* value >0.05). Interestingly, the odds of developing Barrett’s esophagus were significantly lower in those with ESRD versus those without ESRD (OR 0.81 95% CI: 0.716–0.925, *p* = 0.002) (Table 2 and Figure 2).

## 4. Discussion

The more significant results from this study were the increased incidence and risk of GERD in those with CKD compared to those without CKD. Additionally, the highest risk and incidence of GERD were found in the early stage of CKD and tended to decrease as renal function worsened. This is the first study using extensive inpatient patient data to specifically investigate GERD in CKD patients, whose upper GI symptoms might often be misinterpreted. This is consistent with several prospective studies in ESRD patients. Kawaguchi et al. reported that the GERD prevalence in chronic renal failure subjects requiring hemodialysis in Japan was significantly more elevated than in the general Japanese population (24.2% vs. 16.3%) [22]. In another study conducted by the same group of researchers in Japan, in 156 late-stage CKD patients who underwent endoscopic examination, GERD prevalence was 34%, and the prevalence was much higher in patients requiring hemodialysis (50%) [15]. A similar finding was reported by Abdulrahman et al., who found that the prevalence of GERD in ESRD patients was almost doubled compared to the control group (77% vs. 38%). They also found that high serum creatinine and no evidence of *H. pylori* infection were significantly associated with GERD in ESRD patients [17].

Importantly, we also adjusted for the risk factors of GERD and CKD in this study. Identifying the various risk factors associated with GERD is crucial in developing effective prevention and treatment strategies [21]. While age, obesity, hiatal hernia, and smoking are some of the most well-known risk factors for GERD, it is important to consider other potential risk factors as well. Studies investigating the risk factors of GERD should adjust for these known risk factors to identify other potential risk factors that may be specific to certain populations or circumstances. By adjusting for these known risk factors, researchers can focus on identifying new and emerging risk factors that may contribute to the development of GERD. Ultimately, a better understanding of the risk factors associated with GERD can help healthcare professionals to provide targeted interventions to prevent or manage this condition, leading to improved outcomes and quality of life for individuals affected by GERD.

Importantly, we have also shown that several GERD-associated complications affect CKD patients more commonly than subjects with no history of CKD, including esophageal stricture and Barrett’s esophagus. While we could still not demonstrate the exact pathophysiology mechanism between CKD and these complications, there was a clear association between them. This study revealed that the incidence and risk of complications associated with GERD were higher in the early stages of the disease. However, the exact mechanism for this observation remains unclear, and further research is necessary to confirm the findings. It is possible that the small sample size of patients in the late stages of CKD with or without esophageal stricture or Barrett’s esophagus could be a factor in this observation. There were no statistically significant differences in the adjusted odds found in CKD stage 4 and 5 (Figure 2), indicating that further retrospective studies using larger databases are necessary to resolve this issue. Another possible explanation for this observation is that in late-stage CKD patients, acid reflux symptoms might not be the predominant issue; instead, symptoms from volume overload, acidosis, and reduced gastric emptying could take precedence, potentially causing an underestimation of GERD incidence. This finding is consistent with our study’s observation that the risk of Barrett’s esophagus was significantly lower in ESRD patients compared to normal patients, despite ESRD patients experiencing more severe symptoms. Therefore, it is important to conduct further research to better understand the mechanism behind the higher incidence and risk of complications in early-stage GERD. This could help in the development of effective interventions and treatment plans for patients with GERD, especially those with CKD. 

The mechanism is still unknown; however, it could be associated with subjects with late-stage CKD, commonly accompanied by severe uremic syndromes requiring more aggressive anti-reflux treatments. These findings indicate that GERD-CKD patients might require frequent monitoring and management for GERD complications. 

The pathophysiology mechanism of how individuals with CKD have greater chance of developing GERD and its complications is still unclear. Impaired renal function-related uremia, fluid overload, and electrolyte abnormality were proposed as the main contributors. However, our findings suggest that late-stage CKD patients, who suffer from uremic toxin accumulation and fluid overload, were not associated with an increased incidence and risk of GERD, as with early-stage CKD patients. This is consistent with the study conducted by Karahan et al., who suggested that no significant difference was found between pre-dialysis and dialysis patients in upper GI symptoms [16]. This indicates that the increased incidence and risk of GERD in CKD patients may be due to some other factors that have not yet been identified besides uremic and fluid overload. These factors may include, but are not limited to, CKD-induced hormonal changes, endocrine disorders, and medication use. 

One of the common complications of CKD is hyperparathyroidism, which is induced by hypocalcemia and phosphate accumulation [23]. Interestingly, Alexandra et al. found that all symptoms of GERD improved after parathyroidectomy [24]. A similar finding was observed in another two years of prospective study after parathyroidectomy. James et al. found that both motility and acid reflux symptoms significantly improved after parathyroidectomy [25]. Additionally, increased lower esophageal sphincter pressure and the relief of GERD symptoms were observed in patients after parathyroidectomy [26]. Vitamin D deficiency-induced hypocalcemia is another common complication in CKD patients [23]. Interestingly, hypocalcemia was associated with increased LES pressure, whereas calcium infusion, which mimics hypercalcemia in healthy volunteers, decreased LES pressure [26]. This finding may be another possible explanation for our conclusion that GERD prevalence decreased in late-stage CKD patients compared to early-stage CKD patients. As renal function deteriorates in CKD patients, worsening hypocalcemia could increase LES pressure and ameliorate GERD. 

Moreover, CKD is associated with the continued activation of the renin–angiotensin–aldosterone system (RAAS), which was found to play a part in the pathophysiology of GERD. A study performed by Eleonora et al. demonstrated that some RAAS components, such as angiotensin-converting enzyme (ACE) and angiotensin 1 receptor (AT1R), were significantly altered in individuals diagnosed with GERD [27]. GERD has also been shown to be related to the cytokine-mediated pathway [28,29]. Cytokines and oxidative stress have been increasingly recognized as essential contributors to CKD [23]. It has been suggested that oxidative stress and cytokine accumulation in CKD are essential in early inflammatory changes in patients with GERD [29]. Yoshida’s group found that several inflammation markers, such as interleukin-6 and interleukin-8, were significantly higher within esophageal biopsy samples from GERD patients [30,31]. Oxidative stress, which was dysregulated secondary to renal insufficiency, was also involved in the pathogenesis of GERD [29]. It has been suggested that the accumulation of oxidative stress could cause LES relaxation, mucosal injury, and advancement into Barrett’s esophagus [32]. 

Overdiagnosis and underdiagnosis of GERD are important issues in current patient populations. Overdiagnosis can lead to unnecessary testing, treatment, and costs, while underdiagnosis can result in untreated symptoms, complications, and reduced quality of life [33]. Overdiagnosis of GERD can occur when patients are diagnosed with the condition based solely on their symptoms without undergoing diagnostic testing. This can lead to inappropriate treatment, such as the long-term use of proton pump inhibitors (PPIs), which can have potential side effects such as bone fractures, Clostridioides difficile infections, and pneumonia. On the other hand, underdiagnosis of GERD can occur when patients have atypical symptoms, or when diagnostic testing is not performed due to limited access to endoscopy or pH monitoring. Several studies have investigated the rates of the overdiagnosis and underdiagnosis of GERD in different patient populations. For example, a study of patients referred for upper endoscopy found that nearly half of the patients with typical GERD symptoms did not have esophagitis or other endoscopic findings suggestive of GERD. Another study found that up to 70% of patients with reflux symptoms did not have GERD based on 24 h pH monitoring. To address the issue of overdiagnosis and underdiagnosis of GERD, some experts have called for a more standardized approach to the diagnosis and management of the condition. This may include the use of validated symptom questionnaires, more selective use of diagnostic testing, and a greater emphasis on lifestyle modifications in addition to medication therapy.

Various limits were recognized in this study. One of the main limitations of a national inpatient database retrospective study is that the data collected may not be complete or accurate. Hospital records may not contain all relevant information about a patient, and coding errors can occur, which may lead to inaccurate or incomplete data. For example, a patient’s medical history may not be recorded, which could affect the interpretation of the study’s results. Another limitation of a national inpatient database retrospective study is that it can be difficult to establish causality. Since the study is observational, we cannot manipulate the exposure or the outcome, and therefore cannot definitively establish cause and effect. Instead, we can only identify associations between the exposure and the outcome. Additionally, a national inpatient database retrospective study may be limited by selection bias. The data collected may only include patients who were admitted to hospitals, which could exclude those who were treated in outpatient settings. This could affect the generalizability of the study’s findings to the broader population. Finally, a national inpatient database retrospective study may be limited by confounding variables. There may be other factors that influence the relationship between the exposure and the outcome, which the researchers may not have accounted for. This could lead to spurious associations between the exposure and the outcome. It is important to note that the use of ICD-10 codes to identify patients with certain diagnoses may not always reflect the real diagnosis and can potentially underestimate the number of cases. While these codes are useful for standardization and communication between healthcare providers, they are not always comprehensive and can lead to errors in diagnosis and reporting.

Furthermore, the diagnosis of GERD and each complication assessed in this study was centered upon ICD-10-CM codes, which were entered from several hospital groups and electronic medical records. Additionally, the assumption is that the GERD complications, including esophageal stricture and Barratt’s esophagus, were diagnosed based on images, endoscopy, and pathology. The risk factors of GERD, such as smoking, diabetes, and hiatal hernia, were determined via ICD-10-CM codes as well; no timeline for these risk factors could be identified. Finally, ICD-10-CM codes were used to identify subjects with different stages of CKD, which is assumed to be diagnosed based on the glomerular filtration rate (GFR).

## 5. Conclusions

In conclusion, this study highlights the importance of the early detection and management of GERD in patients with CKD, even in the early stages of the disease. The findings suggest that patients with CKD and esophageal symptoms should be thoroughly evaluated for GERD and related complications, as prompt treatment with anti-acid medications may prevent the development of more serious complications. It is well-established that GERD is a common complication in patients with CKD, with a higher prevalence in advanced stages of the disease. However, this study emphasizes the need to consider GERD as a potential complication in patients with early-stage CKD who present with esophageal symptoms, as timely intervention may prevent disease progression and improve outcomes. The clinical implications of this study are significant, as it provides clinicians with a rationale to actively screen and manage GERD in patients with CKD. Early diagnosis and treatment of GERD may not only prevent the development of more serious complications, but may also improve quality of life for these patients. In addition to anti-acid medications, lifestyle modifications such as weight loss, dietary changes, and elevating the head of the bed may also be recommended as part of the management of GERD in patients with CKD. Healthcare providers should also closely monitor these patients for potential drug interactions and side effects of medications. The findings from this study emphasize the importance of considering GERD as a potential complication in patients with CKD and esophageal symptoms, even in early stages of the disease. Early intervention with anti-acid medications and lifestyle modifications may prevent the development of more serious complications and improve outcomes for these patients. Further research is needed to better understand the underlying mechanisms of GERD in CKD and to optimize its management.

## Figures and Tables

**Figure 1 jpm-13-00827-f001:**
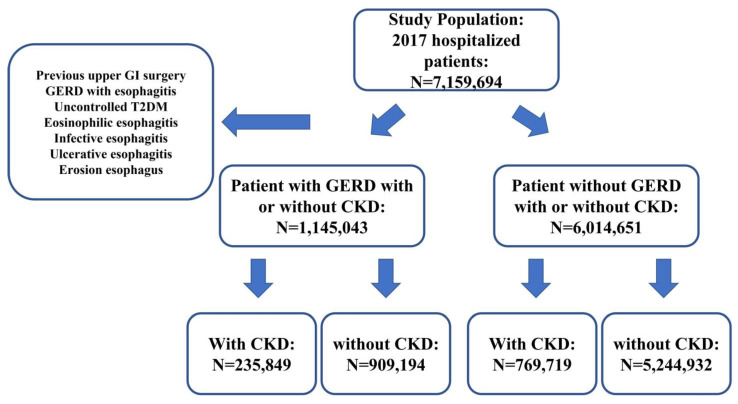
Sample selection and data design flowchart.

**Figure 2 jpm-13-00827-f002:**
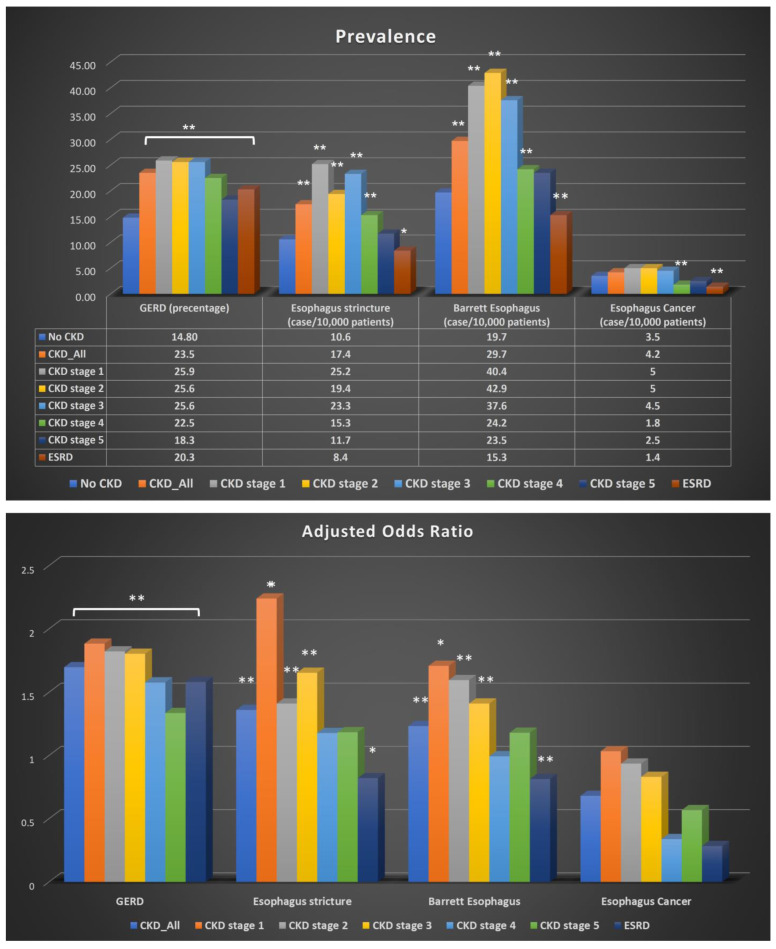
Bar graph of prevalence and odds ratio for CKD patients with GERD or GERD-related complications including Esophagus stricture, Barrett’s Esophagus and Esophagus cancer. Upper, Prevalence of GERD, Esophagus stricture, Barrett’s Esophagus and Esophagus Cancer in patients with or without different stages of CKD. Lower, Adjusted odds ratio of GERD, Esophagus stricture, Barrett’s Esophagus and Esophagus cancer in patients with different stages of CKD. GERD, gastroesophageal reflux disease; CKD, chronic kidney disease; ESRD, end-stage renal disease. Adjusted for age, sex, race, obesity, hiatal hernia and history of smoking. * *p* < 0.05, ** *p* < 0.01.

**Table 1 jpm-13-00827-t001:** Demographics between GERD with and without CKD T2DM, type 2 diabetic mellitus; HTN, hypertension; HLD, hyperlipidemia. GERD, gastroesophageal reflux, disease; CKD, chronic kidney disease.

	GERD w/ CKD	GERD w/o CKD	*p* Value	No GERD w/CKD	No GERD w/o CKD	*p* Value
**Age**	71.6 ± 0.1	62.5 ± 0.1	<0.01	69.9 ± 0.1	43.4 ± 0.1	<0.01
**Sex**						
** Female**	124,690 (52.9%)	546,663 (60.1%)	<0.01	344,777 (44.8%)	3,021,122 (57.6%)	<0.01
** Male**	111,150 (47.1%)	362,502 (39.9%)	<0.01	424,924 (55.2%)	2,222,916 (42.4%)	<0.01
**Race**						
** White**	161,472 (68.5%)	679,222 (74.7%)	<0.01	461,824 (60.0%)	3,142,505 (59.9%)	>0.05
** Black**	41,320 (17.5%)	98,546 (10.8%)	<0.01	161,606 (21.0%)	748,476 (14.3%)	<0.01
** Hispanic**	16,298 (6.9%)	62,590 (6.9%)	<0.01	78,580 (10.2%)	698,702 (13.3%)	<0.01
** Asian**	4377 (1.9%)	13,420 (1.5%)	<0.01	22,083 (2.9%)	173,819 (3.3%)	<0.05
**Complications**						
** T2DM**	108,257 (45.9%)	55,930 (6.2%)	<0.01	362,579 (47.1%)	151,975 (2.9%)	<0.01
** HTN**	123,433 (52.3%)	494,247 (54.4%)	<0.01	336,194 (43.7%)	1,509,476 (28.8%)	<0.01
** HLD**	129,974 (55.1%)	362,456 (39.9%)	<0.01	338,254 (50.4%)	918,667 (17.5%)	<0.01

**Table 2 jpm-13-00827-t002:** The prevalence and odds ratio for GERD and GERD-associated complications in patients with CKD OR, odds ratio; GERD, gastroesophageal reflux, disease; CKD, chronic kidney disease. Adjusted for age, sex, race, hiatal hernia, obesity and smoking.

	**GERD**					
	**GERD Number**	**GERD Percentage**	**Odds Ratio**	***p* Value**	**Adjusted Odds Ratio**	***p* Value**
CKD_ALL			1.768	<0.01	1.697	<0.01
YES	235,849	23.5%				
No	909,149	14.8%				
CKD stage 1	1384	25.9%	2.013	<0.01	1.883	<0.01
CKD stage 2	12,444	25.6%	1.986	<0.01	1.822	<0.01
CKD stage 3	102,156	25.6%	1.968	<0.01	1.803	<0.01
CKD stage 4	24,689	22.5%	1.675	<0.01	1.577	<0.01
CKD stage 5	2675	18.3%	1.293	<0.01	1.336	<0.01
ESRD	41,502	20.3%	1.469	<0.01	1.581	<0.01
	**Esophageal Stricture**					
	**Cases**	**Cases per 100,000 Patients**	**OR**	***p* Value**	**Adjusted OR**	***p* Value**
CKD_ALL			1.642	<0.01	1.36	<0.01
YES	1335	17.4				
No	5547	10.6				
CKD stage 1	10	25.2	2.392	<0.01	2.24	0.012
CKD stage 2	70	19.4	1.834	<0.01	1.411	<0.01
CKD stage 3	691	23.3	2.202	<0.01	1.653	<0.01
CKD stage 4	130	15.3	1.448	<0.01	1.178	0.06
CKD stage 5	14	11.7	1.109	<0.01	1.184	0.531
ESRD	136	8.4	0.789	<0.01	0.821	0.025
	**Barrett’s Esophagus**					
	**Cases**	**Case per 100,000 Patients**	**OR**	***p* Value**	**Adjusted OR**	***p* Value**
CKD_ALL			1.514	<0.01	1.233	<0.01
YES	2288	29.7				
No	10,313	19.7				
CKD stage 1	16	40.4	2.057	<0.01	1.708	0.034
CKD stage 2	18	42.9	2.185	<0.01	1.595	<0.01
CKD stage 3	1117	37.6	1.915	<0.01	1.411	<0.01
CKD stage 4	206	24.2	1.234	<0.01	0.994	0.935
CKD stage 5	28	23.5	1.194	0.35	1.18	0.392
ESRD	250	15.3	0.78	<0.01	0.814	<0.01
	**Esophageal Cancer**					
	**Cases**	**Cases per 100,000 Patients**	**OR**	***p* Value**	**Adjusted OR**	***p* Value**
CKD_ALL			0.848	0.01	0.681	<0.01
YES	272	3.5				
No	2185	4.2				
CKD stage 1	2	5	1.21	0.788	1.034	0.962
CKD stage 2	18	5	1.196	0.451	0.936	0.781
CKD stage 3	135	4.5	1.09	0.33	0.831	0.04
CKD stage 4	15	1.8	0.423	<0.01	0.339	<0.01
CKD stage 5	3	2.5	0.603	0.381	0.568	0.327
ESRD	22	1.4	0.324	<0.01	0.284	<0.01

## Data Availability

The data presented in this study are available on request from the corresponding author. The data are not publicly available due to patient and hospital information privacy and the requirement of H.CUP.

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
