# Peer review of "The Relationship between Gastroesophageal Reflux Disease and Chronic Kidney Disease"

_jpm, 2023, doi:10.3390/jpm13050827_

Round 1

Reviewer 1 Report

Dear editorial team

according to manuscript entitle ' the relationship between gasteroesophageal reflux and chronic renal disease"

1-the authors must mention  the limitations of study

2-how  the authors  match the groups by age sex  and diet as  healthy control  with   or without GERD ?

3  the next prospective  design  research  needed  to solve  the limitations.

Author Response

1-the authors must mention the limitations of study

Response: We have concisely addressed the study's limitations and incorporated them into the end of the discussion section.

2-how  the authors  match the groups by age sex  and diet as  healthy control  with   or without GERD ?

Response: Thank you for your comments. In this retrospective study using the National Inpatient Database, we don't have a traditional "healthy control" group. Our control groups include patients without GERD and CKD, and those without GERD but with CKD. To provide further clarity, we have updated Table 1 with our control groups: "Patients without GERD with CKD" and "Patients without GERD without CKD." Additionally, we have adjusted for age, sex, race, and other covariates across all four groups to ensure comparability.

3  the next prospective  design  research  needed  to solve  the limitations.

Response: We agree with the importance of addressing the limitations concerning the comparison of healthy subjects and GERD patients with or without CKD. To further substantiate our findings, we plan to conduct a prospective study comparing these groups, which will enable us to better understand the impact of GERD on kidney function in both healthy subjects and CKD patients.

Reviewer 2 Report

Thank you for submitting your manuscript for review. I enjoyed reading it and appreciated the somewhat uncommon proposition that CKD may associate with GERD in a mechanism where CKD may be causal instead of the more conventional PPI associated with CKD where PPIs are proposed causal. I think there is merit to exploring this in detail and revision to the manuscript to better support your position. Suggestions for improvement are listed below.

Intro: The diagnosis of GERD is inaccurate. Typically it is dependent on pH monitoring scores and symptom association. GI society guidelines could help you reformulate this description. You could mention ICD codes get entered not using diagnostic criteria used in guidelines if you'd like here.

Methods: Controlling for a co-variate of GERD like hiatal hernia is ill advised. This lesion strongly correlates with the development of GERD and probably does not change risk factor for CKD. It may wash out real GERD when doing this control. A limitation is with ICD based diagnosis code research is in correct diagnosis and controlling for hiatal hernia probably removes the more "real" GERD from the data set. Obesity also probably shouldn't be controlled for due to similar reasons. It may be better to control for comorbid illness not related to the diagnoses of interest. Controlling for generalized illness scores may be better. Even controlling for age can be problematic as I think CKD risk increases with age. 

Figure 1: Missing a box for no GERD and No CKD. This should show how many in total cohort and number excluded total as well. 

Results:

We should assess PPI usage in GERD patients if possible in your data set. Many studies associate PPI regardless of GERD with CKD and perhaps that drives your findings. If the association persists with and without PPI your paper will be MUCH more cited as it would refute many studies linking PPI with CKD instead of GERD. Controlling for PPI in modeling would again wash out true GERD from the dataset, so would consider just comparing GERD/PPI to GERD/no-PPI as far as CKD rates. 

Seems CKD stage is inversely proportional to the risk of GERD. Is this true? If so, why? In association studies if causation is likely should see a dose response curve where more exposure the more of the disease we see but this is not true here. Furthermore, most CKD progresses through stage 1-5/ESRD. Your presented data suggests GERD is "Cured" with progression of CKD through the stages which makes me strongly questions the validity of the findings presented. The pattern seems conserved across the data which suggests it is a real finding, but this needs better explanation. 

Working to decrease wordiness and work toward brevity would help readers for this manuscript. 

Author Response

Thank you for submitting your manuscript for review. I enjoyed reading it and appreciated the somewhat uncommon proposition that CKD may associate with GERD in a mechanism where CKD may be causal instead of the more conventional PPI associated with CKD where PPIs are proposed causal. I think there is merit to exploring this in detail and revision to the manuscript to better support your position. Suggestions for improvement are listed below.

-Thank you for taking the time to review our manuscript. We appreciate your valuable feedback and suggestions.

Intro: The diagnosis of GERD is inaccurate. Typically it is dependent on pH monitoring scores and symptom association. GI society guidelines could help you reformulate this description. You could mention ICD codes get entered not using diagnostic criteria used in guidelines if you'd like here.

Response: Thank you for your valuable recommendation on the diagnosis of GERD. We have revised the introduction section to incorporate the 2022 ACG guidelines, ensuring a more accurate and current depiction of diagnostic criteria. We also acknowledge the potential discrepancies in ICD codes compared to guideline-based diagnostic criteria, providing context for our data source limitations.

Methods: Controlling for a co-variate of GERD like hiatal hernia is ill advised. This lesion strongly correlates with the development of GERD and probably does not change risk factor for CKD. It may wash out real GERD when doing this control. A limitation is with ICD based diagnosis code research is in correct diagnosis and controlling for hiatal hernia probably removes the more "real" GERD from the data set. Obesity also probably shouldn't be controlled for due to similar reasons. It may be better to control for comorbid illness not related to the diagnoses of interest. Controlling for generalized illness scores may be better. Even controlling for age can be problematic as I think CKD risk increases with age. 

Response: Thank you for your insightful comments on the co-variables and potential limitations in our study. We acknowledge the concerns regarding controlling for hiatal hernia, obesity, and age, as these are established risk factors for GERD. While our objective is to investigate the impact of CKD on GERD prevalence and risk, we understand the potential shortcomings of using ICD-based diagnosis codes. In light of your suggestions, we will consider controlling for comorbid illnesses not directly related to the diagnoses of interest, and explore generalized illness scores as an alternative. We also plan to conduct a prospective study with real patients from regional hospitals to validate our findings, addressing the limitations of our current research.

Figure 1: Missing a box for no GERD and No CKD. This should show how many in total cohort and number excluded total as well. 

Response: Thank you for this valuable comments and sorry for this mistake. We add an additional two groups in Table 1.

Results:

We should assess PPI usage in GERD patients if possible in your data set. Many studies associate PPI regardless of GERD with CKD and perhaps that drives your findings. If the association persists with and without PPI your paper will be MUCH more cited as it would refute many studies linking PPI with CKD instead of GERD. Controlling for PPI in modeling would again wash out true GERD from the dataset, so would consider just comparing GERD/PPI to GERD/no-PPI as far as CKD rates. 

Response: We appreciate the comments made by the reviewer. Unfortunately, the medication information was not included in the current database. However, we are planning to conduct a prospective study to address this issue.

Seems CKD stage is inversely proportional to the risk of GERD. Is this true? If so, why? In association studies if causation is likely should see a dose response curve where more exposure the more of the disease we see but this is not true here. Furthermore, most CKD progresses through stage 1-5/ESRD. Your presented data suggests GERD is "Cured" with progression of CKD through the stages which makes me strongly questions the validity of the findings presented. The pattern seems conserved across the data which suggests it is a real finding, but this needs better explanation. 

Response: Thank you for raising this intriguing point. Our study has indeed observed the highest incidence and risk of GERD in the early stages of CKD, with a decreasing trend as CKD progresses. Although no statistically significant differences were found between each CKD stage and ESRD, there could be several reasons for this observation.

First, as CKD progresses, the number of cases may decrease, which could lead to an underestimation of GERD incidence and risk. Second, in late-stage CKD patients, acid reflux symptoms might not be the predominant issue; instead, symptoms from volume overload, acidosis, and reduced gastric emptying could take precedence, potentially causing an underestimation of GERD incidence.

While it is possible that this finding is genuine, we acknowledge the need for further prospective studies to confirm it. We have provided a potential explanation in the discussion section (lines 194-201) to address this matter.

Round 2

Reviewer 2 Report

While improved, I still believe the findings must be tempered. This is cross-sectional study, not adjusted for PPI usage (known to be associated with CKD in other similar studies) and we see the association of GERD decrease as CKD progresses to ESRD. This would seem to suggest the findings are not causal (implied causality typically follows a dose response - more dose more disease).  if CKD was associated, the progression of CKD is associated with decreasing prevalence of GERD which would be atypical as few patients with GERD. Few patients resolve with time, in fact, most get worse as esophageal motility decreases and weight increases with age. None the less, I think the limitation of the study is the way it has to be done in this data set and we probably just have to take it as is or not. I like the novelty of considering CKD as the source of GERD relative to the larger PPI literature.